# Future Orientation and Symptoms of Anxiety and Depression in Italian University Students during the COVID-19 Pandemic: The Role of Resilience and the Perceived Threat of COVID-19

**DOI:** 10.3390/healthcare10060974

**Published:** 2022-05-24

**Authors:** Rocco Servidio, Costanza Scaffidi Abbate, Angela Costabile, Stefano Boca

**Affiliations:** 1Department of Cultures, Education and Society, University of Calabria, 87036 Rende, Italy; a.costabile@unical.it; 2Department of Psychology, Educational Science and Human Movement, University of Palermo, 90128 Palermo, Italy; costanza.scaffidi@unipa.it (C.S.A.); stefano.boca@unipa.it (S.B.)

**Keywords:** resilience, future orientation, depression, anxiety, COVID-19 disease, university students

## Abstract

Several studies have already examined the psychological impact of COVID-19 on psychological well-being in samples of the general population. However, given the importance of future orientation for university students’ mental well-being, it is important to explore whether and how COVID-19 affected this vulnerable population. Therefore, the current study aims to investigate the protective role of future orientation and the mediating effect of resilience on negative emotional symptoms (anxiety and depression). An online questionnaire was administered to a sample of 244 non-infected Italian university students. The mediation analysis results indicated that resilience fully mediated the relationship between anxiety and depression. Finally, the perceived threat of death from COVID-19 moderated the association between resilience and anxiety. This study shows that university students with higher levels of future orientation exhibit higher resilience levels and, consequently, fewer symptoms of negative emotions.

## 1. Introduction

In the last two years, the COVID-19 pandemic has caused much distress to people’s lives, either directly or indirectly, through the effects of the non-pharmacological measures adopted to contain the spread of the virus among the population. The impact of the pandemic on psychological well-being has recently been explored in the general population. In a study conducted in April 2020, using a probability sample (*N* = 1468) and the Kessler-6 psychological distress scale, 13.6% of US adults reported symptoms of severe psychological distress, compared to 3.9% in 2018 [1]. In another study among 9565 individuals from 78 countries, taken at the height of the lockdown, the pandemic was experienced as at least reasonably stressful by most people, with 11% reporting the highest levels of stress.

COVID-19 has had an impact on the lives of college students. With the spread of this infectious disease in February 2020, most universities decided to adopt distance learning approaches to stop viral transmission in the classrooms [2,3,4]. Consequently, the majority of university students decided to leave their campus, thus experiencing uncertainty and worries about academic deadlines, as well as a lack of social support, and often feeling uncomfortable with the new educational approaches based on distance learning [5,6]. This kind of loss can be overwhelming, and university students who lack appropriate coping strategies and social support may experience psychological distress [7,8]. Indeed, previous studies found that similar socially challenging and personal transitions are indeed stressful for university students and are systematically associated with depression and anxiety [9,10,11,12]. Given the consequences of maturational changes, the fact of entering a new social community or new life-changing social roles, and the decline in social support, university students may be seen as more vulnerable to psychological conditions [13]. However, the self-efficacy theory [14] suggests that confident people can cope with future adverse events and show fewer maladaptive symptoms. Individuals with a high level of future orientation should therefore be able to cope with stressful life events better.

According to [10], future orientation can be conceptualized as an individual’s subjective view of his or her personal future. It is based on the human skills to predict the future and anticipate representations and projects, and it represents one of the essential characteristics of human beings [14]. Therefore, future orientation can be considered a multidimensional cognitive motivational construct that provides the core for setting future goals and plans [15,16,17] and developing expectations and personal meaning in relation to future events [18]. For this reason, future orientation has been regarded in the literature as a protecting factor that helps to prevent problematic behaviours [19], as well as to perform adaptive behaviours [20]. The hypothesis of a protecting role of future orientation has been recently tested [21,22,23,24,25], finding a relationship between future orientation, on the one hand, and the reduction of drug use, the avoidance of risky sexual behaviour, and the avoidance of involvement in violent situations, on the other [23]. At the same time, some results have highlighted the positive consequences of future orientation for successful professional outcomes [26].

The situation determined by the spread of the COVID-19 pandemic must be regarded as a generalized stressful event, potentially affecting people’s social life and psychological functioning [27]. The importance of exploring the risk of psychological distress among university students derives from the fact that the current sanitary situation due to COVID-19 requires an active coping effort [28]. Previous studies suggest that those who score high in future orientation perform better and are less distressed by social isolation, loneliness, and free movement restrictions [19,29]. University students who systematically think about their future and who have a strong drive towards personal success are more inclined to fulfil their plans and are less anxious, since they feel less stress from the COVID-19 restrictions [30]. Additionally, students who feel more responsible for their future are more inclined to increase their learning efforts and to achieve better academic results than students who avoid thinking about their careers [29,31]. 

Therefore, the present study investigated the relationships between future orientation and negative emotional symptoms of anxiety and depression in Italian students who had never contracted the COVID-19 infection before completing the survey. Thus, it was expected that people who regularly think about their future may face everyday life problems by preparing in advance and dealing successfully with possible obstacles [23].

Previous studies have demonstrated that individuals do not experience only negative emotions during stressful situations, like the COVID-19 epidemic, but also positive ones [24,32]. Positive emotions can be bolstered by increasing personal resources such as resilience. Resilience is conceptualized as the process of an individual’s adaptability to face adversity and the ability to “bounce back” from stressful experiences [33,34]. Thanks to the “self-regulation mechanism”, human organisms can regulate their interaction with environmental stimuli and continuously adjust to any changes. Cross-sectional and longitudinal studies have demonstrated that resilience has a mediated impact on depressive symptoms and family functioning [8,34,35]. Thus, resilience might protect individuals against negative emotions by decreasing or neutralizing the adverse impact of risk factors of depression and, therefore, help people deal with negative emotions [36].

Given the enduring nature of the COVID-19 pandemic and its widespread and long-lasting effects on mental health, there is a need to clarify the role of those factors (such as resilience) that may protect against the onset of anxiety and depression [32], especially among vulnerable populations such as university students. The present study hypothesizes that future orientation can reduce the adverse effects of negative emotional symptoms such as anxiety and depression by increasing resilience. 

Theoretically, resilience refers to an interactive dynamic construct, an upbeat coping style, and a positive personality trait [37] that promotes more positive psychosocial outcomes during challenging situations such as the COVID-19 pandemic. Notably, the process model of mental resilience suggests that protective factors either mobilize or reintegrate the personal resources needed to cope with life events. To the best of our knowledge, the role of resilience as a mediator in the relationships between future orientation and depressive or anxiety symptoms among Italian university students during the COVID-19 pandemic has not been examined yet. Therefore, based on a survey of the existing literature, we hypothesized that future orientation would be positively related to resilience (H1) and negatively related to maladaptive symptoms (anxiety, H2a, and depression, H2b), respectively. Resilience, in turn, would work as a mediator in these relationships (H3).

Although a previous study had explored the mediating role of resilience in the relationship between future orientation and anxiety [30], there was a lack of research analysing the potential moderator role of fear induced by the spread of COVID-19. Functional levels of stress and fear as adaptive reactions are natural and useful for facing new social challenges such as the pandemic situation, insofar as they promote protective behaviours (maintaining physical distance, washing one’s hands, etc.) that facilitate COVID-19 prevention [38]. By contrast, insufficient or severe levels of fear may be maladaptive and detrimental to physical and mental health [39]. We therefore assumed that the perceived threat of death from COVID-19 could moderate (H4) the associations between future orientation, resilience, and negative emotional symptoms (anxiety and depression). Therefore, it was important not only to examine the moderating effects of COVID-19 risks on psychological well-being but also to learn how moderator variables might reduce psychological distress and increase positive psychological resources [40,41].

## 2. Materials and Methods

### 2.1. Participants and Procedure

An initial sample of 251 participants was recruited online during regular teaching activities. At the end of the data cleaning procedures, the final sample included 244 Italian students (60 males (24.60%) and 184 females (75.40%)) attending various university degrees courses. The participants’ ages ranged from 18 to 40 years (*M* = 22.71 years, *SD* = 3.75). Most participants were attending psychological and educational courses (47.10%). The rest were enrolled in various courses such as economics (13.50%), life science (29.90%), medicine (5.30%), and the humanities (4.10%).

All participants were invited to fill in an anonymous online survey, which was handed out between March and October 2021. They all volunteered for the study, and none received any reward. Moreover, they were also allowed to withdraw their data from the study at any stage. Completing the online questionnaire took approximately 15 min. The language of the questionnaire was Italian. All the research materials and procedures were designed according to the guidelines laid out by Ethics in Human Research and the Italian Association of Psychology.

### 2.2. Measures

The online survey included socio-demographic questions about participants’ gender, age, and degree course.

Negative emotional states were assessed using the Italian version of the Depression, Anxiety, and Stress Scale-21 (DASS-21) [42]. This scale is a self-report instrument consisting of three 7-item subscales and designed to assess a person’s level of depression, anxiety, and stress over the past week. For the present study, we used only the two subscales for depression (e.g., *I felt that I had nothing to look forward to*; α = 0.90) and anxiety (e.g., *I felt I was close to panic*; α = 0.84). The responses were given on a 4-point Likert scale, ranging from 0 (Does not apply to me at all) to 3 (Applies to me most of the time), with higher scores indicating a more negative experience in the past week. 

Future orientation and resilience were measured through the Italian version of the Design My Future scale [43]. This scale includes 19 items rated on a 5-point Likert-type scale from 1 (It describes me not at all) to 5 (It describes me very well). The future orientation subscale (e.g., Building a positive future for myself is something that I think about often; α = 0.93) and resilience subscale (e.g., I believe in achieving my goals; α = 0.82) in the present study exhibited good reliability.

The perceived threat of death from COVID-19 was a single-item measure adapted from Norris et al. (2006). The single item was formulated by asking, “Have you feared that you might die from the coronavirus?” The answer was rated on a binary scale: “Yes”, “No”.

### 2.3. Data Analyses

We used SPSS 26 software to run the preliminary statistical analyses. First, we performed descriptive statistics and Pearson correlation analyses on the research variables of the study. Gender and age were included as covariates in all the statistical analyses to check the effects on anxiety and depression [11]. Given the properties of the constructs, solutions based on item parcelling rather than on individual items are more appropriate to reduce the risk of convergent problems and improve model fits [44,45]. The item parcelling was generated by applying a balanced procedure designed to combine high and low inter-correlation values [46]. Therefore, we consistently used three indicators for each of the four constructs for future orientation and resilience and two indicators for anxiety and depression, respectively. Then, as the first step in the structural equation model (SEM), the measurement model for the latent constructs was tested. Furthermore, we tested the full (measurement and structural) model. In the last step, to test the study’s hypotheses, mediation and moderation analyses were conducted using Mplus 7.01 [47]. The models were estimated with the maximum likelihood parameter, with standard errors and a mean-adjusted chi-square test statistic robust to non-normality (MLM). The MLM chi-square test statistic is also referred to as the Satorra–Bentler (S-B) chi-square. The fit of the tested models was assessed using the following multiple indexes: (a) a comparative fit index (CFI) ≥ 95, (b) a Tucker–Lewis Index (TLI) ≥ 95, (c) a root mean square error of approximation (RMSEA) ≤ 06, and d) a standardized root mean square residual (SRMR) < 08 [48]. Finally, a multigroup moderation analysis was performed, and comparisons were made by computing χ^2^ difference (Δχ^2^).

## 3. Results

### 3.1. Correlation among the Variables

Table 1 shows the outcomes of the descriptive and the Pearson correlations. Negative correlations emerged between future orientation and anxiety, *r*(244) = −0.28, *p* < 0.001, and depression, *r*(244) = −0.50, *p* < 0.001. A significant positive association was observed between resilience and future orientation, *r*(244) = 0.60, *p* < 0.001. Finally, the perceived threat of death from COVID-19 was positively and significantly associated with anxiety, *r*(244) = 0.24, *p* < 0.001. Since all the correlations between the variables were significant, this result satisfied the conditions for performing the subsequent analyses.

### 3.2. SEM and Mediation

The results of the measurement model including future orientation, resilience, anxiety, and depression fit well with the data, robust χ^2^(29, *N* = 244) = 48.64, *p* = 0.013, CFI = 0.98, TLI = 0.97, RMSEA = 0.05, 90% CI [0.02, 0.08], SRMR = 0.04. Since the measurement model results were good, we modelled the effects among the latent variables to test the hypotheses of the study. The results of the SEM analysis (measurement and structural model combined) are shown in Figure 1. The tested model, controlled for age and gender, fit the data well, robust χ^2^(45, *N* = 244) = 85.63, *p* < 0.001, CFI = 0.96, TLI = 0.95, RMSEA = 0.06, 90% CI [0.04, 0.08], SRMR = 0.05.

The results reported in Figure 1 suggest that future orientation has a direct and positive effect on resilience, β = 0.68, *p* < 0.001. In turn, resilience has a negative effect on anxiety, β = −0.34, *p* < 0.01, and on depression, β = −0.45, *p* < 0.001. Moreover, no significant direct effects emerged between future orientation and negative emotional symptoms (anxiety and depression). Resilience fully mediated the relationships between future orientation and anxiety, β = −0.14, SE = 0.05, *t* = −2.71, *p* < 0.01 of the total effect, β = −0.11, SE = 0.04, *t* = −2.49, *p* < 0.05, as well as between future orientation and depression, β = −0.30, SE = 0.06, *t* = −4.71, *p* < 0.01, of the total effect β = −0.46, SE = 0.06, *t* = −7.35, *p* < 0.001.

### 3.3. Moderation

We also explored whether the perceived threat of death from COVID-19 moderated the effects of future orientation on resilience, anxiety, and depression, respectively. Therefore, we performed a multigroup comparison. The model depicted in Figure 1 was tested simultaneously for participants who indicated “No” and ones who indicated “Yes” in relation to fear of death from COVID-19. This first step allowed us to obtain a baseline model, whose fit was acceptable, robust χ^2^(102, N = 244) = 166.01, *p* < 0.001, CFI = 0.94, TLI = 0.92, RMSEA = 0.07, 90% CI [0.05, 0.09], SRMR = 0.07. Then, the equivalence of the measurement model across groups was tested. Results indicated that factor loadings were equivalent across the two groups of participants, Δχ^2^(5) = 7.59, *p* = 0.18. Next, a series of comparisons were performed to detect differences across the two groups by testing single parameters. As shown in Figure 1, the only difference that emerged was between resilience and anxiety, Δχ^2^(1) = 4.75, *p* < 0.05: this relationship tended to be weaker when participants claimed to experience “No” fear of death from COVID-19, β = −0.27, *p* < 0.05, than when they answered “Yes”, β = −0.28, *p* < 0.01.

## 4. Discussion

The restrictions adopted to contain the spread of the virus and the uncertainty associated with the pandemic-related social emergency may have caused a further increase in mental health risks and problems among university students transitioning through this vulnerable life phase [5,8,11]. The present study aimed to explore the mediating role of resilience in the relationship between future orientation and maladaptive emotions (anxiety and depression) and the moderating effect of the perceived threat of death from COVID-19 on the relationship between resilience, on the one hand, and anxiety and depression, on the other, in a sample of non-infected Italian university students. Specifically, we examined the contribution of specific protective factors such as future orientation and resilience, as well as the moderating role of fear of death, to better understand how these variables can identify groups at risks of psychological distress, like university students, and thus to contribute to designing specific intervention strategies. The current results show that resilience fully mediates the relationships between future orientation and anxiety and depression, while the perceived threat of death from COVID-19 moderates the association between resilience and anxiety but not the relationship between resilience and depression.

We found a direct association between future orientation and resilience (H1 was supported). Therefore, the current sample of Italian university students with future orientation goals is more resilient, indicating that they exhibit more positive attitudes toward life and are better equipped to cope with anxiety and depression. The results of the current study are in line with the literature, showing that positive attitudes that outweigh negative thinking can protect against depression and anxiety, as shown by previous studies [13,15,35]. Indeed, a positive outlook on the future is one of the main characteristics of resilience since it helps university students to develop a positive view of life [8]. As indicated in a previous study, positive future orientation became functional during the pandemic, allowing university students to shift their focus from stressful events to the anticipation of future happiness, revealing their ability to apply helpful coping strategies to reduce both anxiety and depression [30].

We found that resilience was negatively correlated with anxiety (H2a supported) and depression (H2b supported). In line with a large body of previous investigations, more subjective well-being resources and effective coping strategies are associated with low levels of depression and anxiety [12,34]. In particular, the results of this study indicate that future orientation reduces anxiety and depression through the mediating role of resilience (H3 was supported). Therefore, the current sample of Italian university students with high future orientation perspectives shows greater motivation to accomplish their life plans by adopting adaptive coping strategies. Additionally, since resilience works as an adaptive coping mechanism that makes individuals better fitted for life, this strategy leads them to experience fewer negative emotions such as anxiety and depression.

Further findings of this study showed that the perceived threat of death from COVID-19 moderated the relationship between anxiety and resilience. University students who experienced fear of death from COVID-19 showed a higher positive relationship between resilience and anxiety (H4 partially supported). This result is consistent with a recent study indicating that COVID-19 risk perception enhances voluntary health-promoting behaviour [39]. However, no other significant moderating effects were found.

This research made it possible to monitor the psychological well-being of the university students and predict who would experience psychological distress. Therefore, the current results may contribute to the definition of intervention programmes aimed at helping university students recover from epidemic-generated distress, and also to cope with future challenges.

## 5. Limitations and Future Directions

The study has some limitations that open up new opportunities for future investigations. First, the research sample is not very large, so the next step should be to expand the research to a more representative sample of college students. The very fact that many of our respondents were enrolled in Psychology or Education courses might be seen as undermining the validity of the study, which is why a more balanced cohort would be desirable in future investigations. Second, a more robust measure of those constructs that have been repeatedly associated with well-being and successful adaptation, such as a positive evaluation of oneself, of one’s own life, and of the future, could overcome the weaknesses inherent in self-report measures. In this regard, [17] performed a lexical study to identify a set of positive orientation markers and used them in an implicit association test meant to assess implicit positive orientation. Another point that requires further investigation is the moderating role of the perceived threat of death from COVID-19. The divergence between those who fear dying from COVID-19 and those who do not, albeit significant, is very small. This may be due to a small significant effect or to our inability to assess the perceived threat of death. This point should be addressed in future investigations. The results of the current study also suggest relevant implications from a practical perspective. For example, it would be worth examining whether making future orientation salient for an individual could provide a motivational drive that increases behaviours contributing to the achievement of the very goals and plans at the core of the individual’s expectations for his or her future.

## Figures and Tables

**Figure 1 healthcare-10-00974-f001:**
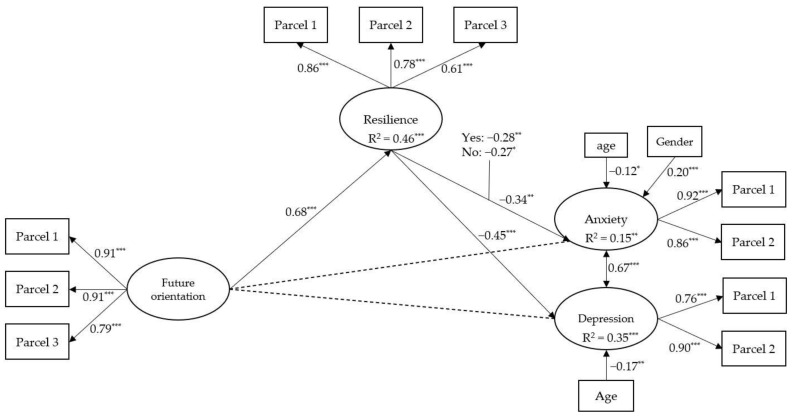
Results of the SEM model. All the values are standardized. The dashed lines indicate non-significant paths. Latent factors are presented in the circle; measured variables (parcels) are presented in the rectangles. All the analyses were controlled for gender (1 = male, 2 = female) and age. Note. * *p* < 0.05. ** *p* < 0.01. *** *p* < 0.001.

**Table 1 healthcare-10-00974-t001:** Descriptive statistics and correlations.

Variable	*M*	*SD*	Skewness	Kurtosis	1	2	3	4	5	6	7
1. Future orientation	3.26	0.64	0.11	0.07	-						
2. Resilience	3.50	0.79	−0.28	−0.21	0.60 ***	-					
3. Anxiety	1.88	0.61	0.78	0.42	−0.28 ***	−0.14 *	-				
4. Depression	2.26	0.73	0.32	−0.63	−0.50 ***	−0.39 ***	0.60 ***	-			
5. COVID-19 threat	-	-	-	-	−0.05	0.04	0.24 ***	0.11	-		
6. Age	22.71	3.75	1.87	5.05	0.15 *	−0.03	−0.14 *	−0.20 **	0.12	-	
7. Gender	-	-	-	-	0.58	0.08	0.15 *	0.00	0.12	0.01	-

Note. Gender (1 = male, 2 = female) and COVID-19 perceived threat of death (0 = no, 1 = yes) are point serial correlations (*r*_pb_). * *p* < 0.05. ** *p* < 0.01. *** *p* < 0.001.

## Data Availability

The data presented in this study are available on request from the corresponding author.

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
