# Peer review of "Future Orientation and Symptoms of Anxiety and Depression in Italian University Students during the COVID-19 Pandemic: The Role of Resilience and the Perceived Threat of COVID-19"

_healthcare, 2022, doi:10.3390/healthcare10060974_

Round 1

Reviewer 1 Report

This is a well-presented paper, elegant and concise. The only shortcomings that I can detect that the authors have not themselves acknowledge are are the imbalance in gender and the prevalence of psychology and education students that filled out the questionnaire. Psychology students are often the lab rats and in this instance, may well be better equiped to understand resilience and other behavioural processes associated with surviving adversity. A more balanced cohort covering more disciplines would strengthen the work.

Author Response

Response to Reviewer 1 Comments

We want to thank the Reviewer 1 for their appreciated feedback. Thank you very much for your timely and very useful review of our paper. We know that reviewing takes time, and we are thankful for the amendments. We have revised our manuscript based on these comments and queries. Significant changes are highlighted in red colour in the main text. We believe our manuscript is now more robust due to the reviewers’ comments.

Thank you for the feedback on our paper. In the following, we will answer each point raised by the reviewer.

This is a well-presented paper, elegant and concise. The only shortcomings that I can detect that the authors have not themselves acknowledge are the imbalance in gender and the prevalence of psychology and education students that filled out the questionnaire.

Psychology students are often the lab rats and, in this instance, may well be better equiped to understand resilience and other behavioural processes associated with surviving adversity. A more balanced cohort covering more disciplines would strengthen the work.

Authors’ response: Thanks for this feedback. We have now highlighted this aspect in the limitation section.

Reviewer 2 Report

Thanks for the opportunity to review the article, “Future orientation and symptoms of anxiety and depression in Italian university students during the COVID-19 pandemic: The role of resilience and the perceived threat of COVID-19.” The general idea of the study is interesting but I had some concerns regarding the literature review and implications of the research. Please see my comments below:

Abstract

The abstract says, “However, far less is known about the effects of COVID-19 on university students, a population shown to be psychologically vulnerable.” There has been quite a bit of research on the effects of COVID-19 on university students within the last few years. Please update this to best reflect the current literature.

Introduction:

Pg. 1; Lines 26-28; This is too bold of a statement, “In the last two years, the COVID-19 pandemic has brought much distress to everybody’s life, either directly or indirectly through the effects of the non-pharmacological measures adopted to contain the spread of the infection among the population.”

It needs a supporting citation and one cannot assume that non-pharmacological measures have brought distress to “everybody’s life”. Some people liked different aspects of these measures. The supporting literature underneath also makes it clear that it did not impact “everybody’s life”.

Pg. 1; Line 37 Change “all” universities to “most” or “many” universities decided to adopt distance learning. Again, there is no way to prove that “all” universities took this approach.

Pg. 1; Line 38-39 clarify what is meant by “the majority of university students decided to leave their campus” is that in China, given the reference? Or does that refer to students living in dorms? The language utilized does not adequately represent the research being discussed.

Pg. 2; Lines 54-55; Cite the following sentence, “For this reason, future orientation has been described in the literature as a protecting factor able to prevent problematic behaviours.”. Does the research say it can “prevent” problematic behaviours or reduce them?

Pg. 2; Lines 61-63; Remove “all aspects” from the sentence so that it says “potentially affecting people’s social lives and psychological functioning? Be sure to cite this research as well.

Pg. 2; Lines 72; Define what “non-infected” means

Pg. 3; Lines 109-110; Please check formatting guidelines if direct quotes need a page number

Pg. 3; Participants; How did you take into consideration whether or not your sample of college students was more or less prone to depression and anxiety regardless of the pandemic?

In the introduction share more about why you are studying the perceived threat of death of COVID-19 among non-infected college students? Is this a common fear that college students have based on existing literature? How does studying this variable help us better understand the experiences of college students? It feels like this variable was added on late and is less clear how it connects with the literature review and the purpose of the paper.

Discussion

Continue to build out the discussion section. The purpose of the present study was repeated 2-3 times in lines 255-267. Why is it important to study these variables? How specifically do they contribute to existing research?

The purpose of adding on whether COVID-19 moderates the relationship between anxiety and resilience is less clear. Connect this better to the existing literature on why this variable was studied.

Implications

Please share a few implications for this research. How does the findings help researchers, colleges, college students, etc.?

I hope the feedback provided helps improve the quality of your manuscript.

Author Response

Response to Reviewer 2 Comments

We want to thank the Reviewer 2 for their appreciated feedback. Thank you very much for your timely and very useful review of our paper. We know that reviewing takes time, and we are thankful for the amendments. We have revised our manuscript based on these comments and queries. Significant changes are highlighted in red colour in the main text. We believe our manuscript is now more robust due to the reviewers’ comments.

Thank you for the feedback on our paper. In the following, we will answer each point raised by the reviewer.

Thanks for the opportunity to review the article, “Future orientation and symptoms of anxiety and depression in Italian university students during the COVID-19 pandemic: The role of resilience and the perceived threat of COVID-19.” The general idea of the study is interesting but I had some concerns regarding the literature review and implications of the research. Please see my comments below:

Abstract

The abstract says, “However, far less is known about the effects of COVID-19 on university students, a population shown to be psychologically vulnerable.” There has been quite a bit of research on the effects of COVID-19 on university students within the last few years. Please update this to best reflect the current literature.

Authors’ response: Thanks for this feedback. We have now changed the sentence.

Introduction:

Pg. 1; Lines 26-28; This is too bold of a statement, “In the last two years, the COVID-19 pandemic has brought much distress to everybody’s life, either directly or indirectly through the effects of the non-pharmacological measures adopted to contain the spread of the infection among the population.”

It needs a supporting citation and one cannot assume that non-pharmacological measures have brought distress to “everybody’s life”. Some people liked different aspects of these measures. The supporting literature underneath also makes it clear that it did not impact “everybody’s life”.

Authors’ response: Thanks for this suggestion. We have now modified all the critical sentences in the introduction.

Pg. 1; Line 37 Change “all” universities to “most” or “many” universities decided to adopt distance learning. Again, there is no way to prove that “all” universities took this approach.

Authors’ response: Thanks for this suggestion. “All universities” has been changed to “most universities”.

Pg. 1; Line 38-39 clarify what is meant by “the majority of university students decided to leave their campus” is that in China, given the reference? Or does that refer to students living in dorms? The language utilized does not adequately represent the research being discussed.

Authors’ response: Thanks for this suggestion. We have now included the following reference: https://journals.sagepub.com/doi/full/10.1177/00986283211043924

Pg. 2; Lines 54-55; Cite the following sentence, “For this reason, future orientation has been described in the literature as a protecting factor able to prevent problematic behaviours.”. Does the research say it can “prevent” problematic behaviours or reduce them?

Authors’ response: Thanks for this suggestion. We have now clarified this critical sentence by including the following references: doi:10.1002/pchj.283 and doi:10.1111/jopy.12242

Pg. 2; Lines 61-63; Remove “all aspects” from the sentence so that it says “potentially affecting people’s social lives and psychological functioning? Be sure to cite this research as well.

Authors’ response: Thanks for this suggestion. We have now included two references: doi: 10.1371/journal.pone.0240146 and doi:10.1016/j.jad.2021.02.040

Pg. 2; Lines 72; Define what “non-infected” means.

Authors’ response: Thanks for this suggestion. We have now included the following sentence: “Italian students who never contracted Covid before completing the survey.”

Pg. 3; Lines 109-110; Please check formatting guidelines if direct quotes need a page number.

Authors’ response: Thanks for this suggestion. We have now improved the text of the introduction.

Pg. 3; Participants; How did you take into consideration whether or not your sample of college students was more or less prone to depression and anxiety regardless of the pandemic?

Authors’ response: Thanks for this comment. However, we have no reason to think that these students were depressed or anxious ex-ante.

In the introduction share more about why you are studying the perceived threat of death of COVID-19 among non-infected college students? Is this a common fear that college students have based on existing literature? How does studying this variable help us better understand the experiences of college students? It feels like this variable was added on late and is less clear how it connects with the literature review and the purpose of the paper.

Authors’ response: Thanks for this suggestion. We have now clarified the role of the mentioned variables, also by providing new references: doi:10.3389/fpsyg.2021.643977 and doi:10.12688/f1000research.109575.1

Discussion

Continue to build out the discussion section. The purpose of the present study was repeated 2-3 times in lines 255-267.

Why is it important to study these variables? How specifically do they contribute to existing research?

The purpose of adding on whether COVID-19 moderates the relationship between anxiety and resilience is less clear. Connect this better to the existing literature on why this variable was studied.

Authors’ response: Thanks for this suggestion. We have now changed the critical sentence (p. 6): “Specifically, we examined the contribution of specific protective factors such as future orientation and resilience, as well as the moderating role of fear of death, to better understand how these variables can identify groups at risks of psychological distress, like university students, and thus to contribute to designing specific intervention strategies.”

Implications

Please share a few implications for this research. How does the findings help researchers, colleges, college students, etc.?

Authors’ response: Thanks for this suggestion. We have now included a potential practical implication of the study (7): “This research made it possible to monitor the psychological well-being of the university students and predict who would experience psychological distress. Therefore, the current results may contribute to the definition of intervention programs aimed at recovering from the epidemic generated distress in university students but also in future challenges originated from yet unknown events”.

I hope the feedback provided helps improve the quality of your manuscript.

Authors’ response: Thank you for all the useful suggestions.

Reviewer 3 Report

First of all, I would like to congratulate the authors for the paper.
The methodology and the results found are really fantastic, but I find some aspects that need to be improved.
The introduction in some cases looks like conclusions of the study. For example between lines 65 and 70. Furthermore, I do not understand how university students can anticipate a never experienced event such as COVID-19 and the reasons why university students are more vulnerable are not specified.
The sample is really a very important limitation. In addition to being small, it is too heterogeneous. The university or universities from which the sample is drawn are not known, so it is not possible to determine whether it is representative or not.
The discussion hardly compares the results with other studies. I recommend comparing results with those of other authors.
Otherwise, congratulations on your work.

Author Response

We want to thank the Reviewer for their appreciated feedback. Thank you very much for your timely and very useful review of our paper. We know that reviewing takes time, and we are thankful for the amendments. We have revised our manuscript based on these comments and queries. Significant changes are highlighted in red colour in the main text. We believe our manuscript is now more robust due to the reviewers’ comments.

Thank you for the feedback on our paper. In the following, we will answer each point raised by the reviewer.

First of all, I would like to congratulate the authors for the paper. The methodology and the results found are really fantastic, but I find some aspects that need to be improved.

The introduction in some cases looks like conclusions of the study. For example between lines 65 and 70.

Authors’ response: Thanks for this suggestion. We have now revised all the section introduction.

Furthermore, I do not understand how university students can anticipate a never experienced event such as COVID-19 and the reasons why university students are more vulnerable are not specified.

Authors’ response: Thanks for this feedback. We have now clarified why university students are more vulnerable (pp. 1-2). Two additional references have been included, also by considering the Italian context: doi:10.1080/13548506.2021.1891266 and doi:10.3390/ijerph16162864.

The sample is really a very important limitation. In addition to being small, it is too heterogeneous. The university or universities from which the sample is drawn are not known, so it is not possible to determine whether it is representative or not.

Authors’ response: Thanks for this feedback. We have now highlighted this aspect in the section limitation. 

The discussion hardly compares the results with other studies. I recommend comparing the results with those of other authors.

Authors’ response: Thanks for this suggestion. We have now revised all the discussions.

Otherwise, congratulations on your work.

Authors’ response: Thank you for all the useful suggestions.

Round 2

Reviewer 2 Report

Thanks for the updated revisions to your manuscript.